# Current States and Future Trends in Safety Research of Construction Personnel: A Quantitative Analysis Based on Social Network Approach

**DOI:** 10.3390/ijerph18030883

**Published:** 2021-01-20

**Authors:** Xiangcheng Meng, Alan H. S. Chan

**Affiliations:** School of System Engineering and Engineering Management, City University of Hong Kong, Hong Kong 999077, China; alan.chan@cityu.edu.hk

**Keywords:** construction industry, safety research, social network, main path analysis, literature review

## Abstract

The construction industry is recognized as a high-risk industry given that safety accidents and personnel injuries frequently occur. This study provided a systematic and quantitative review of existing research achievements by conducting social network approach to identify current states and future trends for the occupational safety of construction personnel. A total of 250 peer-reviewed articles were collected to examine the research on safety issues of workers in construction industry. Social network approach was applied to analyze the interrelationship among authors, keywords, and citations of these articles using VOS viewer and CitNetExplorer. A knowledge structure map was drawn using main path analysis (MPA) towards the collected papers, which was implemented by Pajek. In line with the findings of social network analysis, five research groups, and six keyword themes were identified in accordance with the times of cooperation of researchers and correlation among keywords of the papers. Core papers were identified by using main path analysis for each research domain to represent the key process and backbone for the corresponding area. Based on the finding of the research, significant implications and insights in terms of current research status and further research trends were provided for the scholars, thus helping generate a targeted development plan for occupational safety in construction industry.

## 1. Introduction

The construction industry is plagued by a high level of occupational risks and has been recognized as a hazardous industry in which fatal and nonfatal occupational injuries frequently occur [1,2,3,4]. For instance, 11.7 out of every 100,000 construction workers in the United States suffered fatal injuries in large-scale construction projects in 2018 [5]. In Hong Kong, the number of industrial fatalities for the construction industry increased from 12 in 2016 to 22 in 2017, which indicated an 83.3% increase [3]. These data indicate the necessity of reducing the occurrence of construction accidents as well as improving the safety of construction personnel. Given the serious problems of personnel safety in construction industry, the studies on the safety issue of construction workers were essential as a vital component of high production efficiency and operational profit of the construction industry [6,7,8]. Those academic achievements were reviewed time to time by scholars in qualitative way (traditional literature review) for examining the safety development of construction personnel [9,10,11,12]. By contrast, the quantitative review on construction safety issues still remains uncommon and insufficient. Compared with the traditional literature review that mainly based on self-judgment and subjective determination, the quantitative literature review has superior objectivity and preciseness for identifying current research status and prospecting further research orientation because the results are derived from data and verified by statistical and computerized approaches [13,14]. Therefore, the lack of quantitative review on construction safety research causes the research gap that hinders the understanding of academic achievements and the development of further research due to the unavoidable subjectivity caused by traditional literature review, which further leads to reduced pertinence and orientation of the identified research status and further research trends. Therefore, this study was conducted to fill this particular research gap by reviewing and identifying the current status and understanding the research prospects of personnel safety in construction industry by using quantitative approaches.

Social network analysis (SNA) was applied as an effective approach for identifying the research trends, domains and opportunities for particular industries by quantitatively reviewing relevant papers. It has been recommended by many scholars as an effective method for citation-based research, which can monitor scientific evaluation and development trends, as well as predict future research directions [15,16]. Compared with the traditional method of literature study, SNA contains less subjectivity and is more suitable for objective analysis and presentation of a visual network layout for the correlation among specific features of articles, such as keywords, citation relationship, and coauthor occurrence, to identify the various research domains and the dynamics of research trend [13,14]. In this study, four typical literature-based networks were formed and analyzed to identify the correlations between authors, keywords, citations, and research main paths. Those four methods were considered effective for the quantitative literature review of construction safety research as they played a significant role in discovering the dynamic procedure of academic knowledge and identifying the information development for decades [17,18,19,20,21]. Specifically, co-authorship analysis was applied since it reveals the speed and quality of scientific cooperation and academic exchanges [17,18], keyword co-occurrence analysis was used as it helps examine the internal correlation among studies with similar keywords [19], citation network analysis was conducted because it provides the graphical presentation of the knowledge increment and clusters the studies in categories [20], and main path analysis was adopted due to its effectiveness of revealing the key advancement of knowledge development about a particular research field [21].

In line with the findings of the present research, the current states and future trends in safety research of construction personnel can be objectively clarified by quantitative approach of literature review. The research aims of this study were to (a) identify the research domain of personnel safety of construction industry; (b) reveal further research opportunities for the safety issue of construction personnel; and (c) provide specific recommendations and suggestions in terms of the further research of personnel safety for construction industry in accordance with the findings.

## 2. Materials and Methods

### 2.1. Literature Collection

A total of 250 sample papers were carefully collected in full-text version from the Web of Science, Google Scholar, and Scopus databases by searching the keywords “safety performance”, “safety improvement”, and “construction worker safety”. The combination of using three common databases can reduce the research bias and improve the reliability of the collected data [22]. Sample size was set at 250 papers to ensure the unbiasedness and reliability of the systematic review of the literature [23]. After the particular keywords were entered, the top 250 papers ranked by research engine, which indicated high relevance between the articles and the keywords, were selected. The extracted articles mainly came from eight peer-reviewed SCI publications and one conference publication, namely, Safety Science (30 papers), Journal of Construction Engineering and Management (36 papers), Automation in Construction (13 papers), Accident Analysis and Prevention (11 papers), Construction Research Congress 2016: Old and New Construction Technologies Converge in Historic San Juan (nine papers), Journal of Safety Research (eight papers), Journal of Management in Engineering (eight papers), Procedia Engineering (PR; six papers), and the International Journal of Environmental Research and Public Health (five papers). The remaining literatures belong to other journals. All the collected papers were published from 1 January 1990 to 31 April 2020.

### 2.2. Social Network Analysis

In this research, the citation-based network analysis was conducted by using VOSviewer (developed by Nees Jan van Eck and Ludo Waltman, Centre for Science and Technology Studies, Leiden University, The Netherlands), CitNetExplorer (developed by Nees Jan van Eck and Ludo Waltman, Centre for Science and Technology Studies, Leiden University, The Netherlands), and Pajek (developed by Andrej Mrvar and Vladimir Batagelj, University of Ljubljana, Slovenia), which are all widely used non-commercial software in literature network analysis. The Appendix A provides details of the algorithm used in the software. 

Four typical networks were formed and analyzed. Namely, the co-author network was established to study the practical forms of scientific collaboration. All aspects of scientific research cooperation can be tracked reliably by analyzing the network of co-authorship [17,18]. A keyword co-occurrence network was established to identify keywords and themes that simultaneously appear in papers, which presented correlations between particular research topic or research direction [19]. The citation network was verified by classifying the papers into different domains to summarize the important research issues, structures, and paradigms [24]. Each research domain was further analyzed through main path analysis (MPA) on Pajek to identify the key process of knowledge development and research trends, thereby pointing out future research opportunity of the safety research in construction industry [25].

## 3. Results and Discussions

### 3.1. Co-Authorship Network

For this study, a total of 595 authors were involved in 250 papers. A threshold was set by minimizing the number of documents of an author as more than two, so as to remove those unrepresentative researchers. The authors whose name only shows once in all 250 articles were discarded from data analysis [26]. Lastly, 44 authors were reserved to construct the co-authorship network. After conducting co-authorship analysis, all 44 authors were classified into five clusters. The classification details are depicted in Table 1.

Figure 1 presents that 11 authors were involved in cluster 1, and six authors were included in cluster 5. A connection existed between clusters 1 and 5, indicating that they were correlated due to the cooperation of researchers. In cluster 1, Li Heng was the core researcher because the node size representing him was the largest among all the nodes in the network. It indicated that he frequently worked with others more and, thus, had the strongest correlation. The major research topic for Li and his coworkers was monitoring and modeling of the safety behavior of construction workers. For cluster 5, the linkage distribution of each node was relatively even compared with that of cluster 1, which indicates that all six authors cooperated with one another equally. The connection between clusters 1 and 5 was contributed by the research cooperation between Zuo and Luo about the impact of attitudinal ambivalence on safety behavior [27].

Figure 2 depicts that the network of cluster 4 consisted of eight authors, among which Chan was the most important author due to his frequent collaborations with other writers. Chan’s recent academic achievements indicate his research interest in the correlation analysis among different safety-related factors, such as the relationship between safety behavior and safety climate for construction workers. The safety perception and hazard recognition of construction workers were also the research interests of Chan and his coworkers.

For cluster 2, Figure 3 illustrates that Fang Dongping worked most frequently with other scholars. Fang’s research interest was mainly related to the modeling of safety management between workers and supervisors. The cause of unsafe work behavior was recently investigated by Fang with the integration of safety climate [28]. The interrelation between clusters 2 and 6 was achieved due to the cooperation between Choudhry and Zahoor on the research in advantages and disadvantages of current safety practices within the theory of safety benchmarking [29].

Figure 4 displays the results of the co-authorship network for cluster 3, in which the most active researcher among all nine authors was Albert, who has collaborated with all the remaining scholars except Rajendran. Eyeball tracking and visual simulation of construction workers were the most preferred topics of Albert in the past few years based on his research outcomes. Risk recognition and perception were also discussed and modeled by Albert with the integration of various influence factors and safety training.

### 3.2. Keyword Co-Occurrence Network

The co-occurrence of author’s keywords constituted the network linkage among different nodes. A total of 605 author-specified keywords were extracted from 250 papers on the basis of bibliometric data. For further simplification, the threshold of keyword co-occurrence was set as more than four times to remove insignificant correlations and 38 keywords eventually remained [26]. Table 2 gives the cluster classification of all keywords and Figure 5. depicts the result of network analysis. The clusters were numbered from 1 to 5 in accordance with their magnitudes.

Cluster 1 mainly included safety constructs and factors that have recently received much attention from scholars, such as safety climate, safety performance, safety citizenship behavior (SCB), and safety attitude. Keywords, for example, job performance and social exchange, were mainly applied as mediating variables in the correlation analysis among safety concepts. It is worth mentioning that SCB, as a new concept of occupational safety, has been declared a promising research issue by scholars [30,31]. The definition of SCB states that it is related to organizational behavior theory, which is the main topic of cluster 5 (organizational citizenship behavior and organizational identification); other keywords, such as helping and servant leadership, were discussed as the influential factors of organizational citizenship behavior [32,33,34].

The risk and hazard-related concepts (hazard identification and recognition, risk perception and assessment) were the key themes in clusters 2 and 4. Numerous researchers have focused on studies of risk and hazard-related theories and the behavior of construction workers, such as the relationship between hazard recognition and risk perception [35,36]. The influencing factors of hazard recognition were investigated [37], and risk perception was considered a key factor of several safety constructs, such as safety performance and safety culture [38,39]. After perceiving risks while working, risk assessment is required to determine whether the workers can undertake these risks. Gambatese et al. designed a particular system model to assess and mitigate upstream risks [40], whereas Man et al. investigated the behavioral base of whether workers are willing to undertake risks [41].

The keywords for cluster 3 were mainly about safety management in terms of personnel characteristics. For instance, visual simulation and proximity warning system were recognized as effective tools for safety training and management [42,43,44]. Moreover, research on safety management inclined to combine with the characteristics of vulnerable groups, such as minority, elderly, and females (gender issue) [45,46,47,48]. Although cluster 6 involved only four keywords (safety behavior, safety awareness, social identity, and social influence), its extensive correlation with clusters 3, 4, and 1 revealed its significance (i.e., connected with safety perception in cluster 3, safety management in cluster 4, and safety climate in cluster 1). For instance, the significance of construction personnel awareness on work safety and its relationship with safety climate were revealed [28], which provided the linkage between cluster 6 and cluster 1; Wei et al., combined safety awareness and building information modeling technology as a systematic model of safety management, which contributed to the correlation between clusters 6 and 4 [43]; Gyekye and Salminen discussed safety perception of construction workers and provided the connection between clusters 6 and 3 [49].

### 3.3. Citation Network

In information science and bibliometrics, a citation graph (or a citation network) is a directed graph in which each node refers to a document and each edge represents a citation relationship from the current publication to another. In this study, the citation network was drawn on CitNetExplorer, which is a non-commercial software used to analyze and visualize citation correlation [26]. The cluster distribution of all 250 papers is depicted in Figure 6. A total of 185 papers were classified into four clusters which were numbered from 1 to 4 in accordance with their magnitude. Safety climate (cluster 1) was the most popular research domain with 84 articles (33.6%), followed by safety design (cluster 2) with 62 articles (24.8%), safety citizenship behavior (cluster 3) with 29 articles (11.6%), and training on hazard perception and recognition (cluster 4) with 10 articles (4%). The remaining 65 articles were scattered and not included in any cluster, which were identified as unclassified clusters. The rationale and algorism of clustering were described in the Appendix A, which have been verified by Van Eck and Waltman for the rationality and reliability [26].

### 3.4. Main Path Analysis

After the four clusters were determined, the knowledge structure of each cluster was identified. On the basis of the guideline of MPA [25], the citations in each cluster were weighted in line with the follow equation to identify the core papers:(1)$$Weight_{ij}=\frac{TP_{ij}}{TSS_{j}},$$
where *TP_ij_* is the aggregate of paths in network j, including reference i, and *TSS_j_* is the sum of paths between the origins (i.e., an article that does not cite another article) and terminuses (i.e., an article that is not cited by others) in network j. The MPA was conducted by using Pajek 2.05. Readers who are interested in the MPA principles can refer to the relevant literature above [25].

#### 3.4.1. Safety Citizenship Behavior

Safety citizenship behavior is an identified research domain in accordance with citation network analysis. Figure 7 shows the developmental milestones of the knowledge on safety citizenship behavior. This type of altruistic behavior was defined as a specific organizational citizenship behavior that aims to ensure the safety of other team members and achieve the safety target of the organization [50,51]. Didla et al., provided clarifications of the motivators and consequences for employees engaged in this type of behavior [52]. Meanwhile, several scholars have discovered the mediating factor of SCB. For instance, Conchie and Donald investigated the mediating effects of safety-specific trust on the relationship between safety leadership and SCB [53]. Reader et al. further expanded the original mediating factor (trust) to more sophisticated concepts (social exchange and organizational support) and evaluated the causality with SCB [33]. Subsequently, Curcuruto and Griffin conducted a comprehensive study of safety citizenship orientation in terms of the mediator and antecedents of SCB, and extensive variables were involved in this particular study [31]. The previous research reveals a broad orientation of safety citizenship research in safety-related issues. For further research expansion, the present study recommends further SCB research to be combined with particular themes which were identified from the keywords co-occurrence. For instance, safety factors like safety leadership and safety climate can be involved in the SCB research for analyzing the correlations among them. The personal characteristics like age and gender can also be used to test the influence mechanism towards SCB [32].

#### 3.4.2. Safety Design

Safety design was identified as the largest research domain for construction workers’ safety, in which the corresponding cluster involved 84 relevant papers, and the main paths of the knowledge structure included 10 core articles. Figure 8 presents the main knowledge structure of this domain. The designer of construction engineering plays an important role in the project management, and the safety performance of the project also depends on the designers’ decision to a great extent [54]. Gambatese et al., indicated that design professionals are critical in enhancing the safety performance of construction site [55]. Subsequently, Choudhry and Fang involved the adaptability of the safety design in workplace as the influential factors of unsafe work behavior [28].

On this basis, several safety projects were designed by scholars for the safety management of particular construction workers. Gangolells et al., developed a project for construction hazard prevention by reviewing and ranking safety risk level of different construction designs [56]. Tam and Fung focused on the study of workplace safety design of tower crane workers to understand the requirement of executing statutory and non-statutory guidelines [57]. Guo et al. applied game technologies for the design of safety platforms and presented a new and useful solution for the safety of construction operations [58]. Teizer et al. developed a particular safety system based on location tracking and data visualization technologies [59]. After conducting a systematic overview and analysis of the safety management design [60], safety design began focusing on the risk prevention of construction work [44,61]. Therefore, further research is suggested to design a system for safety improvement of a particular type workers. Such as the establishment of a risk perception index is recommended for further studies as a screening tool for personnel safety of crane workers. Additionally, proper design for workloads should be considered for safety improvement of construction workers’ performance [38,39].

#### 3.4.3. Safety Climate

The safety climate of working environment is a promising research domain based on citation network analysis, as presented in Figure 9. Siu et al., indicated the significance of safety climate by revealing its influences on safety performance and safety attitude of construction workers [62]. After one year, safety climate was used as a particular indicator for the safety performance of construction workers with psychological strains as mediators [63]. After that, the measurement of safety climate has attracted considerable attention to scholars [28]. Dov developed a multilevel framework for safety climate that identified the differences between organizational and group levels with separate measurement scales [64], based on which Jiang et al., conducted a multilevel study on the relationship between perceived colleague and safety behavior by using safety climate as a mediator [65]. With further in-depth research, Chen et al., verified the significant effect of safety climate on regional difference of accidents and safety violation rates [66]. However, at that time, the common definition of safety climate still remained unclear after a long period of exploration. Additionally, the application on construction management was still lacking [67]. Therefore, further researches were conducted to address those issue. For instance, Marin et al. further investigated the relationship between safety climate and actual injury rate of workers with involving construction-specified safety management practice [68]. Alruqi et al. (2018) provided a consistent definition for this particular construct and every dimensions of safety climate through meta-analysis [69]. For further safety research, safety climate is recommended to be integrated in the influence model as the organizational influence factors of promising safety construct which was identified using MPA and SNA, such as SCB and safety awareness [28].

#### 3.4.4. Training on Hazard Perception and Recognition

The last research domain was related to safety training of hazard perception and recognition, and the main paths of research development involved seven papers. Figure 10 depicts the core articles that contribute the most to the research progress. The significance of safety training was previously discussed by Dedobbeleer et al. by considering the individual perception of hazard as a main focus in conducting a safety training program [70]. Dai and Goodrum emphasized the risk perception and safety training to obtain the perspective of craft workers on construction productivity [71]. The research on safety training practice became more integrated with the personnel perceptions of construction workers, thereby providing project managers with feasible practical implications in safety training sessions [72]. On this basis, numerous scholars have focused on improving safety training in terms of individual perceptions and insights of construction workers. For instance, Namian et al. and Zuluaga et al. investigated safety training with risk perception and hazard recognition by integrating empirical data from different construction projects in the USA [35,36]. Jeelani et al., evaluated the hazard recognition of construction workers by identifying its potential obstacles when staff inspected the working environment [73]. Meanwhile, Jeelani et al., developed an immersive and personalized safety training environment to improve the effectiveness of safety training outcomes [74]. Further research is recommended to assess the influence of safety training and education on the safety constructs, such as SCB and safety awareness, which were identified as the promising research perspective by SNA in this study [35].

### 3.5. Discussion on Further Research Trends

In accordance with the findings of SNA, a framework of proposing future research trends and directions of construction safety is depicted in Figure 11, which includes the categorized keywords and proposed research domains.

The keywords were classified in four categories in line with the co-occurrence analysis in Section 3.2, namely, safety constructs and factors, risk and hazard-related concepts, safety management in terms of personnel characteristics, as well as the organizational citizenship and identification. Combining with the identified research domains and the core processes of knowledge development revealed in main path analysis, several trends and directions for further construction safety research were proposed:

(1) Further studies are recommended to integrate the theories of multiple research domains, thereby expanding the orientations and scopes of the studies. For instance, the study measuring the influences of risk perception and recognition on the safety citizenship behavior of construction workers is considered promising, while the possible effect of safety climate on risk perception and safety citizenship behavior is also suggested as further research topic [35,53]. In addition, the study on the safety design for construction workers should be encouraged, such as the design of theoretical approach (safety intervention program) and technical tool (wearable protective equipment) for eliminating unsafe behavior and improving the safety performance of construction workers [44,61].

(2) Based on the keywords co-occurrence analysis, personal characteristics are usually involved during the safety management of construction workers, therefore, the integration of personal factors towards four promising research domains should be emphasized. Possible trends and orientations could be ① the influence of demographics (age, gender, education level, etc.) on safety climate and safety citizenship behavior; ② the personal characteristic influencing the effectiveness of training on risk perception and recognition; and ③ the design of safety caring and management for particular types of workers (elderly, females, etc.) [32,45,46,47,48]. As shown in keywords co-occurrence network, BIM and VR can be applied as the auxiliary tools of safety management thus achieving visualization [42,43,44].

(3) This study also encourages to modeling the influence mechanism of the safety factors and constructs (categorized by keywords analysis) on the identified research domains of construction industry. For instance, the influence of safety factors (safety leadership, safety altitude, safety climate, etc.) can be involved in the research on safety design, safety citizenship behavior, risk perception, and safety performance of construction workers [30,71]. As shown in keywords co-occurrence network, a quantitative method like structural equation modelling (SEM) can also be used as an auxiliary tool for testing the influence mechanism among safety constructs [31].

(4) Finally, the findings of keywords analysis highlight the importance of organizational identity and sociality for improving safety status of construction workers, which can inspire the further safety research to discover the roles of organizational collectivity and cohesion in the safety improvement of construction workers [32,33,34]. Possible research trends could be: ① the influences of organizational engagement and social relationship exchange on the promotion of individual risk perception and organizational citizenship behavior of construction workers, and ② the design of effective communication channels among construction workers or between superiors and subordinates for avoiding unsafe behavior.

## 4. Conclusions

This study was conducted to make a systematic and quantitative review of existing research achievements and identify the promising research opportunity of construction worker’s safety using social network analysis. By conducting social network analysis, the interrelationships between authors, keywords, and citation relationships among the literature were identified using VOSviewer and CitNetExplorer, so as to reveal the current research states and promising research domains for further construction workers’ safety issue. The core papers of each research domain were identified with the main path analysis implemented by the Pajek software package, which also indicated the key development progresses in construction safety.

### 4.1. Theoretical and Practical Implications

The significance and innovation of the study were reflected by its theoretical and practical implications. For theoretical implication, this study contributed to the knowledge development of construction workers’ safety through quantitative methodology, which can fulfill the research gap in quantitative literature review of construction safety for its current status and further trends by using an innovative SNA approach. First, five research groups were identified based on researchers’ cooperation frequency. The relatively active authors and significant co-author relationships were also identified in the relevant research interests to determine the development of scientific cooperation and academic exchange. Second, keyword co-occurrence was analyzed to determine the popular research themes. Third, citation correlation was obtained after citation network analysis and the four research domains was found, which can cover all the themes of the included papers. Fourth, the development process and research milestones of each research domain were revealed by using MPA.

With regard to the practical implications, the study proposed some perspectives and possible directions for scholars which are of importance for orientating further research on construction safety. For instance, it is suggested for academic researchers that future research should propose expansions from the single research issues to different academic domains and theories. Additionally, the categorized keywords which include concepts of safety factors, demographic, organizational identity, risk perception, and safety management can be integrated with different research domains. For industrial practitioners, significant implications and innovative insights were provided for the safety management of construction workers. For instance, the identified research domains of workers’ safety climate can guide the further attention of the management of construction enterprises to improve the working conditions of construction workers. Construction companies could improve the safety climate by organizing safety weeks in which different activities, such as safety conferences, site visits, and safety video competitions, can be conducted given the promising influence of safety climate on safety research of construction personnel.

### 4.2. Limitation and Future Research 

This study has several research limitations. First, the assumption and principle were based on whether one paper was cited by another paper, and citation network analysis did not consider the negative citations, which indicated that the papers were cited for criticism. This problem was mentioned by Garfield [75]. However, the influence of the negative citation problem was generally considered slight in citation analysis [76]. Second, there are some exclusions of key researchers and keywords in this study since the papers selected focused on major issues of personnel safety. Further study is recommended to expand the search range of databases and keywords when collecting literatures.

## Figures and Tables

**Figure 1 ijerph-18-00883-f001:**
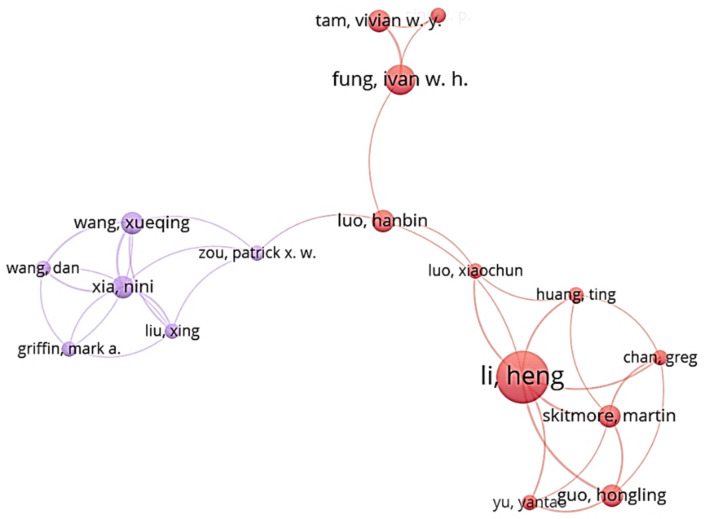
Co-authorship networks of clusters 1 and 5.

**Figure 2 ijerph-18-00883-f002:**
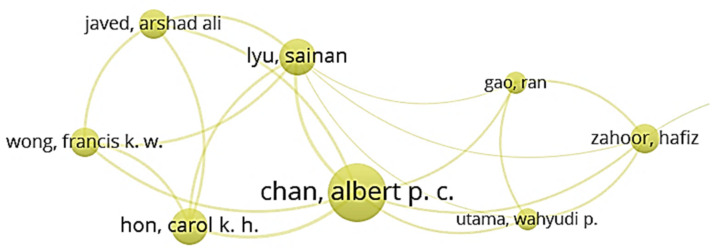
Co-authorship network of cluster 4.

**Figure 3 ijerph-18-00883-f003:**
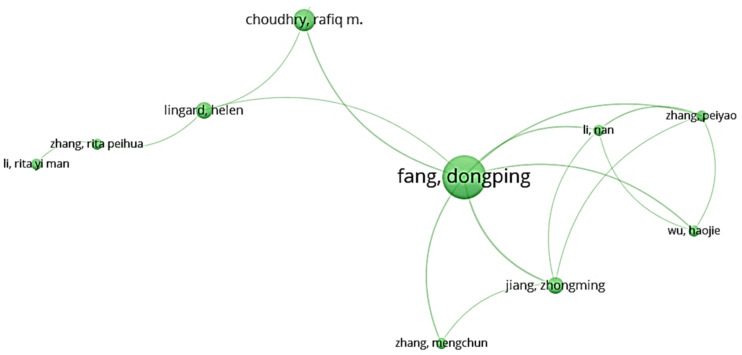
Co-authorship network of cluster 2.

**Figure 4 ijerph-18-00883-f004:**
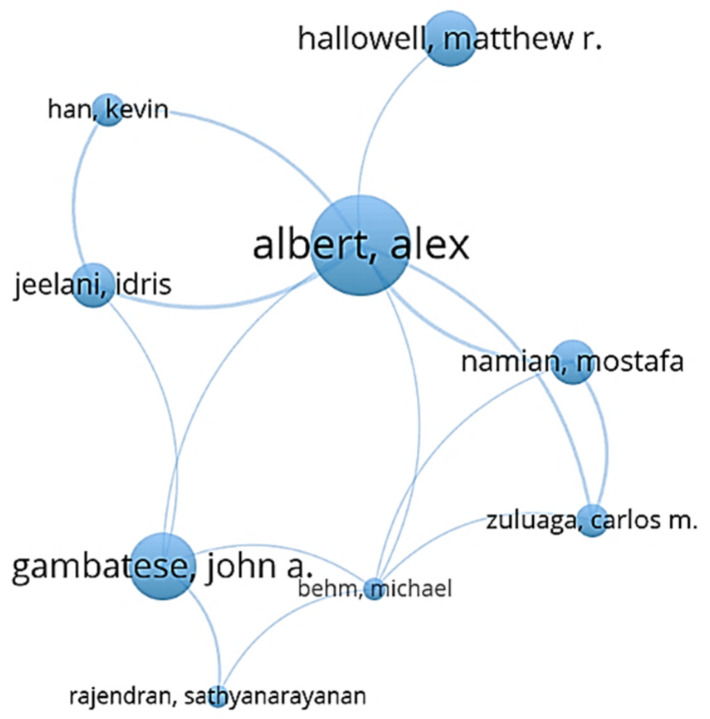
Co-authorship network for cluster 3.

**Figure 5 ijerph-18-00883-f005:**
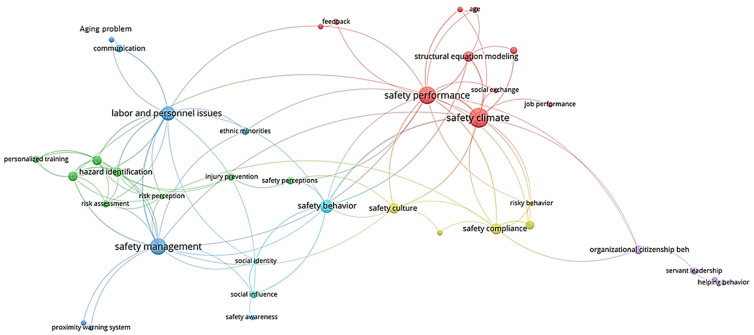
Co-occurrence network for author keywords of collected papers.

**Figure 6 ijerph-18-00883-f006:**
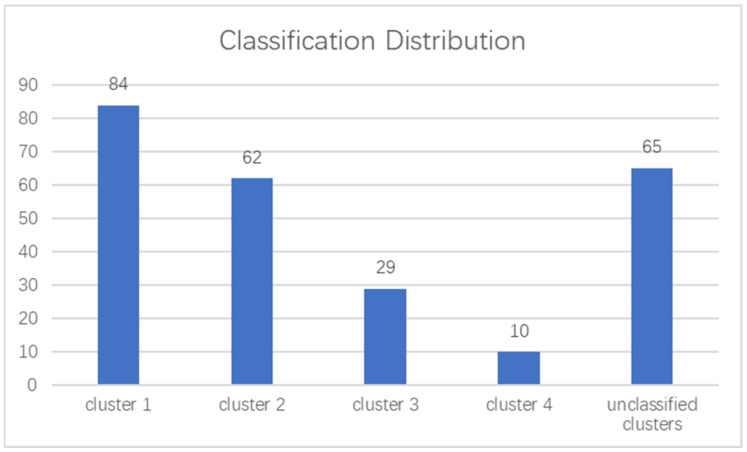
Research domains classification of the literatures.

**Figure 7 ijerph-18-00883-f007:**
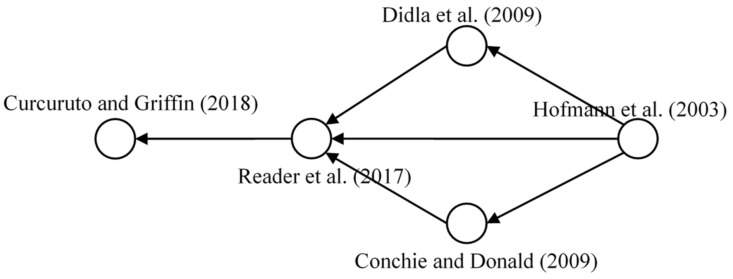
Main path of safety citizenship behavior.

**Figure 8 ijerph-18-00883-f008:**
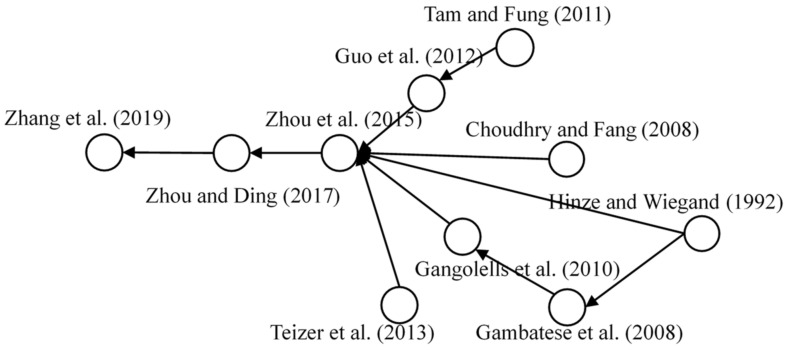
Main path of safety design.

**Figure 9 ijerph-18-00883-f009:**
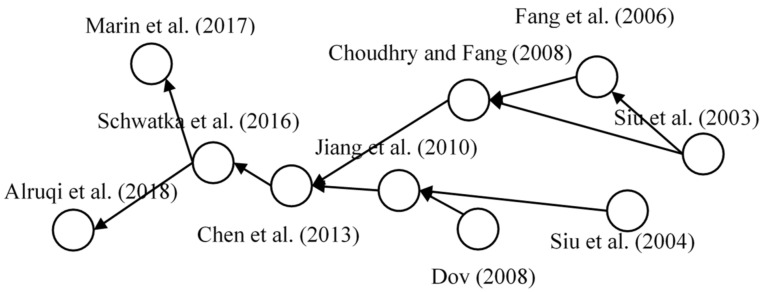
Main path of safety climate.

**Figure 10 ijerph-18-00883-f010:**
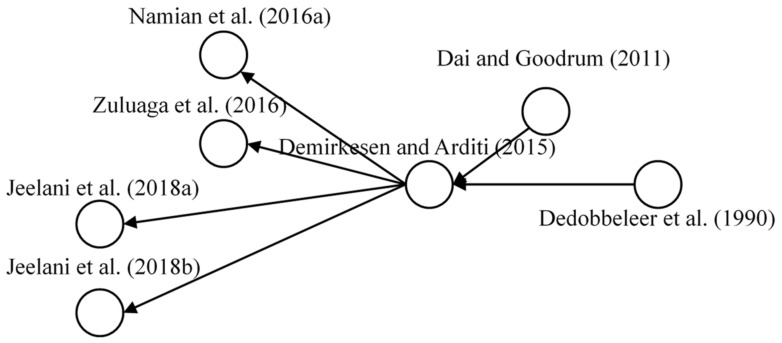
Main path of training of hazard perception and recognition.

**Figure 11 ijerph-18-00883-f011:**
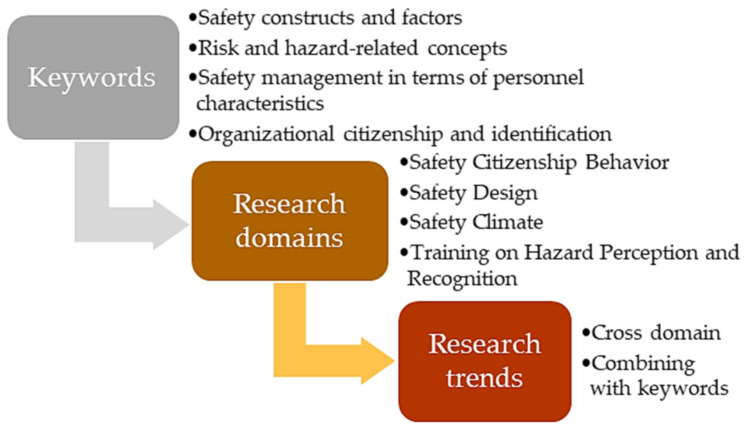
Framework for proposing further research trends and orientation for occupational safety in construction.

**Table 1 ijerph-18-00883-t001:** Cluster classification of authors.

Cluster 1 (Red)	Cluster 2 (Green)	Custer 3 (Blue)	Cluster 4 (Yellow)	Cluster 5 (Purple)
Chan, Greg	Choudhry, Rafiq M.	Albert, Alex	Chan, Albert P. C. (largest node size)	Griffin, Mark A.
Fung, Ivan W. H.	Fang, Dongping (largest node size)	Behm, Michael	Gao, Ran	Liu, Xing
Guo, Hongling	Jiang, Zhongming	Gambatese, John A.	Hon, Carol K. H.	Wang, Dan
Huang, Ting	Li, Nan	Hallowell, Matthew R.	Javed, Arshad Ali	Wang, Xueqing
Li, Heng (largest node size)	Li, Rita Yi Man	Han, Kevin	Lyu, Sainan	Xia, Nini
Luo, Hanbin	Lingard, Helen	Jeelani, Idris	Utama, Wahyudi P.	Zou, Patrick X. W.
Luo, Xiaochun	Wu, Haojie	Namian, Mostafa	Wong, Francis K. W.	
Sing, C. P.	Zhang, Mengchun	Rajendran, SathyanarayanaN.	Zahoor, Hafiz	
Skitmore, Martin	Zhang, Peiyao	Zuluaga, Carlos M.		
Tam, Vivian W. Y.	Zhang, Rita Peihua			
Yu, Yantao				

**Table 2 ijerph-18-00883-t002:** Cluster classification of keywords.

Cluster 1 (Red)	Cluster 2 (Green)	Cluster 3 (Dark Blue)	Cluster 4 (Yellow)	Cluster 5 (Pink)	Cluster 6 (Blue)
Age	Hazard identification	Communication	Risk management	Helping behavior	Safety awareness
Feedback	Hazard recognition	Ethnic minorities	Risky behavior	Organizational citizenship behavior	Safety behavior
Intervention	Injury prevention	Gender	Safety compliance	Organizational identification	Social identity
Job performance	Personalized training	Labor and personnel issues	Safety culture	Servant leadership	Social influence
Safety attitudes	Risk assessment	Aging problem	Safety participation		
Safety citizenship behavior	Risk perception	Safety management			
Safety climate	Safety perceptions	Visualization			
Safety performance	Safety training				
Social exchange					
Structural equation modelling					

## Data Availability

The data presented in this study are available on request from the corresponding author.

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
