# Peer review of "Current States and Future Trends in Safety Research of Construction Personnel: A Quantitative Analysis Based on Social Network Approach"

_ijerph, 2021, doi:10.3390/ijerph18030883_

Round 1

Reviewer 1 Report

This study provided a systematic and quantitative review of existing research achievements by conducting social network approach to identify current states and future trends for the occupational safety of construction personnel. The topic is interesting. However, some comments need to be addressed before publication.

1 why did the authors adopt the SNA method?

2 what are the rational of Cluster classification?

3 why did the authors use the Path Analysis ?

4 the authors are suggested to make the analysis of the cluters in depth, which is too superficial. Then, the authors need to summarize some future trends which can be hepful to the future scholars.

5 what are the innovation of this study?

6 what are the significance of this study?

Author Response

Dear reviewer:

        Thanks so much for your efforts in reviewing this manuscript. all the comments have been addressed and responded in the cover letter, please check it out.

         Yours,

MENG Xiangcheng

Author Response

(The authors gave the same response as above.)
